# Seroprevalence of SARS-CoV-2 in 1186 Equids Presented to a Veterinary Medical Teaching Hospital in California from 2020 to 2022

**DOI:** 10.3390/v14112497

**Published:** 2022-11-11

**Authors:** Kaila Lawton, Stefan M. Keller, Samantha Barnum, Christina Arredondo-Lopez, Kennedy Spann, Nicola Pusterla

**Affiliations:** 1Department of Medicine and Epidemiology, School of Veterinary Medicine, University of California, Davis, CA 95616, USA; 2Department of Pathology, Microbiology, and Immunology, School of Veterinary Medicine, University of California, Davis, CA 95616, USA

**Keywords:** SARS-CoV-2, sick horses, ELISA, prevalence factors

## Abstract

While some companion animals have been shown to be susceptible to SARS-CoV-2, their role in the COVID-19 pandemic has remained poorly investigated. Equids are susceptible to SARS-CoV-2 based on the similarity of the human ACE-2 receptor and reports of infection. Clinical disease and prevalence factors associated with SARS-CoV-2 infection in equids have not yet been investigated. The aim of this study was to determine the seroprevalence of SARS-CoV-2 and selected prevalence factors in 1186 equids presented for various conditions to a Veterinary Medical Teaching Hospital over a two-year period. Blood samples were tested for SARS-CoV-2 antibodies using an ELISA targeting the receptor binding domain (RBD) of the SARS-CoV-2 spike protein. Further, selected prevalence factors (season, age, breed, sex, presenting complaint) were retrieved from the medical records. No information was available on whether the horses had come into contact with COVID-19-positive individuals. Among the study animals, 42/1186 (3.5%) horses had detectable SARS-CoV-2 antibodies. Amongst the prevalence factors investigated, only seasonality (spring) was associated with a greater frequency of seropositivity to SARS-CoV-2. Horses with medical and surgical complaints were more likely to test seropositive to SARS-CoV-2 compared to horses presented for routine health care procedures, suggesting more frequent and/or longer interactions with individuals with COVID-19. While horses can become infected with SARS-CoV-2 via the occasional spillover from COVID-19 individuals, clinical disease expression remains subclinical, making horses an unlikely contributor to the spread of SARS-CoV-2.

## 1. Introduction

Knowledge of animal reservoirs or intermediate hosts of SARS-CoV-2 [1] is continuing to increase with the ongoing COVID-19 pandemic. Understanding which animal species can carry and transmit SARS-CoV-2 is important to control the spread of the virus and to protect vulnerable animal populations, both in the wild and under human care. Research has shown that detecting SARS-CoV-2 infection provides the best evidence for animal susceptibility. Another method used to infer susceptibility in-silico is digital modeling of the angiotensin-I-converting enzyme 2 (ACE-2) structure, a key receptor for the SARS-CoV-2 spike protein [2,3,4,5]. Mutations in the latter protein can enhance its binding affinity for ACE-2 and, thus, facilitate new host adaption and cross-species infection [6]. ACE-2 binding affinity to SARS-CoV-2 has been investigated via protein structure analysis and genome comparison in 410 vertebrate species [2]. Mammal binding affinity to ACE-2 ranged from low to high, with equids categorized as having a low SARS-CoV-2 affinity [2].

Suspected human-to-animal transmission of SARS-CoV-2 has been documented in dogs, cats, ferrets, minks, lions, and tigers, among other animals [7,8]. Horses most likely contract SARS-CoV-2 from humans with clinical and asymptomatic COVID-19 infection [9,10]. Through the course of the pandemic, different SARS-CoV-2 variants [11] have evolved and spread throughout the United States in a series of peaks. The SARS-CoV-2 variants have different levels of contagiousness and pathogenicity, with later variants showing high levels of contagiousness, leading to a greater morbidity rate in humans [11,12,13]. As humans become infected with new SARS-CoV-2 variants, spillover to domestic animals from a COVID-19 individual is more likely to occur. Therefore, it is important to continue to study the impact of potential spillover from humans to animals via longitudinal studies aimed at identifying the frequency of such events and characterizing clinical disease expression. The aim of the present study was to determine the seroprevalence of SARS-CoV-2 in equids presented for various medical and surgical complaints to a veterinary hospital over 2 years and to investigate selected prevalence factors associated with seropositivity to SARS-CoV-2.

## 2. Materials and Methods

### 2.1. Study Population and Sample Collection

Serum and plasma samples were collected from equids presented to the William R. Pritchard Veterinary Medical Teaching Hospital, School of Veterinary Medicine, University of California at Davis. Blood samples were collected from 2 February 2020 to 17 March 2022 with breaks from 1 April 2021 to 7 June 2021 and 9 July 2021 to 9 January 2022. The period of sample collection coincided with the ongoing COVID-19 pandemic. Blood samples were taken as part of the routine diagnostic workups for patients presented to the veterinary hospital for various medical, surgical, reproductive complaints, or routine health care. Whole blood was centrifuged and serum or plasma was stored at −80 °C until analyzed. For the majority of patients, only one blood sample was available; however, horses that were hospitalized for longer periods or horses with repeated follow-up appointments had multiple samples saved. When multiple samples were available, only samples taken 12 days apart were retained, based on the reported median time for seroconversion against SARS-CoV-2 in humans [14]. Records for each patient were reviewed and information pertaining to the date of the visit, signalment (age, sex, and breed), and presenting complaint were recorded. Horses were grouped based on the complaint in one of six broad categories (surgical, orthopedic, reproductive, medical, healthy, and others).

### 2.2. Serology

Antibody detection against SARS-CoV-2 was performed using a previously validated ELISA assay [9,15] which targets the immunodominant receptor binding domain (RBD) of the SARS-CoV-2 spike protein. The cut-off value for positive samples was calculated using six times the standard deviation above the OD mean value of 88 seronegative samples from a pre-COVID-19 group of healthy adult horses [9,16]. A positive control was established from a serum sample from an adult Quarter horse mare, which seroconverted to SARS-CoV-2 post natural infection and confirmed via a plaque reduction neutralization test [10]. Because of the inability to test the serum samples using the reference standard of virus neutralization, seropositive serum samples determined via the ELISA targeting the SARS-CoV-2 RBD of the spike protein were defined as suspect positive.

### 2.3. Statistical Analyses

Descriptive analyses (mean, standard deviation, and median) were performed to evaluate the seasonality, demographics, and presenting complaints. Categorical analyses were performed using Pearson’s chi-square test to determine the association between observations (season, age, breed, sex, presenting complaints) and serological status (suspected seropositive or seronegative against SARS-CoV-2). All statistical analyses were performed using commercial software (Stata Statistical Software, Version 14, College Station, TX, USA) and statistical significance was set at *p* < 0.05.

## 3. Results

A total of 1563 blood samples from 1186 horses were available for serological analysis. A single blood sample was available throughout the study period from 1018 horses, while 2 to 11 samples collected ≥12 days apart were available from 168 horses. A greater number of samples was submitted during the winter season compared to other seasons.

The study population ranged in age from 0 to 45 years with a median of 13 years. Quarter horses, Thoroughbreds, and Warmbloods were the most common breeds represented (Table 1). There were 55.5% males (stallions and geldings) and 41.5% females, while the sex of the remaining 3% of animals was not recorded. The majority of horses were presented for medical or surgical reasons or routine health care, and less frequently for orthopedic, reproductive, and other reasons (Table 1).

A total of 44 blood samples from 42 horses (3.5%) were suspect seropositive for SARS-CoV-2. The ages of the suspected SARS-CoV-2 seropositive horses ranged from 0–30 years, with a median of 12 years. Quarter horses, Thoroughbreds, and Warmbloods were, again, the most common breeds (Table 1). The percentage of males was slightly higher for this group at 61.9% compared to the seronegative study population. Medical complaints were the most frequent reasons for presentation of the suspected SARS-CoV-2 seropositive and seronegative patients (Table 1). Amongst the selected prevalence factors investigated, only seasonality (spring) was associated with a greater frequency of suspect seropositivity to SARS-CoV-2 (*p* < 0.001). Among the suspected SARS-CoV-2 seropositive horses, 8 horses had more than one blood sample tested. In one horse, the serological status switched from suspect SARS-CoV-2 seropositive to seronegative between two samples collected 6 months apart. For the remaining 7 horses, the serological status changed from SARS-CoV-2 seronegative to seropositive with samples collected 13 days to 7 months apart (Figure 1).

## 4. Discussion

Studies have shown that various domestic animal species are susceptible to SARS-CoV-2 infection through natural and/or experimental infection [17,18]. Little is known about the role of domestic livestock species, such as equids, in the SARS-CoV-2 pandemic. Bosco-Lauth et al. evaluated the susceptibility of domestic livestock to SARS-CoV-2 and found that these animals showed no clinical disease, no viral shedding as determined through qPCR of nasal and fecal samples, and no isolation of the virus from respiratory tissues [19]. Further, a 2020 study from China showed no antibodies specific to SARS-CoV-2 from 18 equine serum samples [20], but in 2022, the first known report of SARS-CoV-2 in equids occurred in Thoroughbred racing horses exposed to asymptomatic COVID-19 individuals [9]. In the latter study, horses were determined to have a 5.9% serological rate of suspect SARS-CoV-2 infection [9], which is high considering that equids have been shown to have a low binding affinity of the ACE-2 receptor to SARS-CoV-2 [2]. This high percentage of suspect seropositivity was hypothesized as being due to the large number of COVID-19-infected humans that interacted with these racing horses on a regular basis [9]. All studies reporting on seropositive horses have highlighted the lack of clinical disease, a situation that has also been reported in other domestic and livestock animal species [9,21,22,23,24]. Although these animals had detectable SARS-CoV-2 antibodies, they did not display COVID-19 clinical signs. To the authors’ knowledge, this is the first study determining antibodies against SARS-CoV-2 in a population of horses with various medical and surgical complaints during the COVID-19 pandemic.

Various virus, host and environmental risk factors contribute to the spread of COVID-19 infection in humans, including length of exposure, number of people infected, age, gender, ethnicity, and the presence of a pre-existing medical condition [25]. Similarly, the seroprevalence of SARS-CoV-2 in horses likely relates to a combination of frequency and duration of interactions with clinical or asymptomatic COVID-19 individuals. In humans, it is more common for the elderly to contract clinical COVID-19 [25]; however, a study in cats showed that SARS-CoV-2 replicated more efficiently and showed a higher mortality rate in younger cats [26]. In horses, a previous study documented seropositivity more commonly in younger horses [9,10]. Moreover, 18/42 horses (42.9%) in this study were between 0 and 10 years, suggesting that younger horses are more susceptible to SARS-CoV-2 infection, however, this finding was not statistically significant. Seropositivity has previously been reported in Thoroughbreds and Quarter horses [9,10], showing diversity in equine breed susceptibility. Susceptibility to SARS-CoV-2 does not appear to be associated with breed as our results showed that seropositivity was observed amongst a wide range of different horse breeds. In humans, most positive samples for COVID-19 are from men [25], but a study with dogs and cats showed no association between seropositive cases and the sex of the animal [27]. Although 61.9% of our seropositive cases were geldings and stallions, we also had a higher percentage of overall samples from male animals, resulting in a lack of association between sex and serological SARS-CoV-2 status. COVID-19 is generally categorized as a respiratory disease, but even seropositive horses likely remain subclinical [10]. Among the 42 suspected SARS-CoV-2 seropositive horses, patients with medical and surgical complaints showed a higher seroprevalence compared to animals presented for routine health care procedures. Horses that need more human interactions, whether due to disease or training, may have a greater likelihood of becoming seropositive for SARS-CoV-2. This observation may relate to the fact that patients in need of greater care require longer and more frequent interactions with individuals. Seasonality was shown to be the only demographic category in our research, with a statistically significant increase in seropositive samples during the spring season. COVID-19 infection rates in humans also show strong seasonal patterns, with most cases of new infections being reported during winter and spring [28,29]. The seasonality associated with suspect seropositive horses emphasizes the significance of researching the potential expansion of the viral host range.

Limitations of the study relate to the inability to confirm active SARS-CoV-2 infection through the testing of nasal secretions via qPCR. Further, the SARS-CoV-2 seropositive blood samples were not confirmed through the reference standard of virus neutralization due to the small amount of serum or plasma samples available. These limitations could have impacted true seroprevalence against SARS-CoV-2. The lack of longitudinal data for the majority of the horses represents an additional limitation as only 168 horses had ≥2 samples tested. While the majority of these horses remained seronegative, eight horses had changes in their antibody titers with one seropositive horse reverting to a seronegative status and seven horses experiencing seroconversion. The design of the study cannot determine or differentiate whether SARS-CoV-2 infection occurred from spillover from individuals with COVID-19 or between horses. Lastly, the susceptibility of horses to the different SARS-CoV-2 variants circulating during the study period could not be determined. The main COVID-19 variants in the United States (alpha, delta, and omicron) were at their peaks during April 2021, September 2021, and December 2021 through January 2022, respectively [11,30]. Looking at the seasonal infection rate for the study horses, no noticeable increases in positive cases were observed during the alpha and delta coronavirus variant peaks; however, 11 of 42 suspect seropositive samples (26.2%) were collected during spring 2022. This period coincides with the peak period of the highly contagious omicron coronavirus variant and supports our hypothesis that horses likely contracted SARS-CoV-2 due to spillover from humans.

## 5. Conclusions

Companion animal susceptibility to COVID-19 is a complex interaction of viral, host/recipient, and environmental factors. Present studies suggest that equids are potential dead-end hosts of SARS-CoV-2 due to occasional spillover from individuals with COVID-19. Our results showed that 3.5% of horses presented to the veterinary hospital had detectable antibodies specific to SARS-CoV-2. Horses with medical and surgical complaints were seen to test seropositive to SARS-CoV-2 more frequently than horses presented at the hospital for routine health care suggesting more frequent and/or longer interactions with individuals, some of them having clinical or asymptomatic COVID-19. Seasonal detection of SARS-CoV-2 seropositive horses paralleled the seasonal peak of COVID-19 in human populations. It is important to continue to monitor and test natural and experimental transmission of SARS-CoV-2 between equids and other domestic animals in order to best protect vulnerable populations.

## Figures and Tables

**Figure 1 viruses-14-02497-f001:**
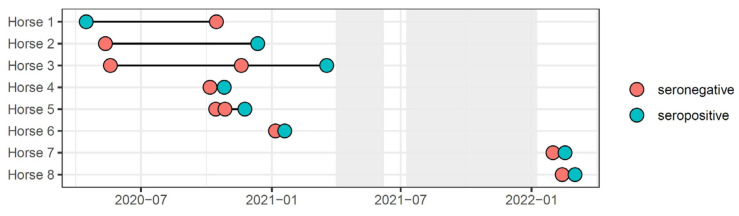
Timeline of blood samples from 8 horses with antibody titer changes against SARS-CoV-2. Individual horses are listed with continuous numbers (Horse 1–8). Temporal ELISA titers against SARS-CoV-2 are expressed as positive (orange circle) or negative (blue circle). The shaded areas represent periods with no sample collections.

**Table 1 viruses-14-02497-t001:** Signalment (age, breed, sex), presenting complaint, and season of presentation in 1186 equids tested for antibodies against SARS-CoV-2.

		Total (1563)	Positive Cases (42)	Negative Cases (1521)
Age		0–45 (median: 13)	0–30 (median: 12)	0–45 (median: 13)
Breed	Quarter Horse	475 (30.39%)	12 (28.57%)	463 (30.44%)
	Thoroughbred	234 (14.97%)	8 (19.04%)	226 (14.86%)
	Warm Blood	157 (10.04%)	5 (11.90%)	152 (10.00%)
	Arabian	89 (5.69%)	1 (2.38%)	88 (5.79%)
	Friesian	40 (2.56%)	1 (2.38%)	39 (2.56%)
	Paint	68 (4.35%)	1 (2.38%)	67 (4.40%)
	Others	500 (31.99%)	14 (33.33%)	486 (31.95%)
Sex	Male ^1^	867 (55.47%)	26 (61.90%)	841 (55.29%)
	Female	648 (41.46%)	16 (38.10%)	632 (41.55%)
	Unknown	48 (3.07%)	0 (0%)	48 (3.16%)
Complaint	Surgical	312 (19.96%)	11 (26.20%)	301 (19.79%)
	Orthopedic	217 (13.88%)	9 (21.43%)	208 (13.68%)
	Reproductive	39 (2.50%)	1 (2.38%)	38 (2.50%)
	Medical	567 (36.28%)	14 (33.33%)	553 (36.36%)
	Healthy	301 (19.26%)	5 (11.90%)	296 (19.46%)
	Others	127 (8.12%)	2 (4.76%)	125 (8.22%)
Season	Spring (Feb-April)	380 (24.31%)	16 (38.10%)	364 (23.93%)
	Summer (May-July)	368 (23.54%)	9 (21.43%)	359 (23.60%)
	Fall (Aug-Oct)	352 (22.52%)	7 (16.67%)	345 (22.68%)
	Winter (Nov-Jan)	463 (29.62%)	10 (23.81%)	453 (29.78%)

^1^ Represents geldings and stallions.

## Data Availability

Not applicable.

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
