# Peer review of "Seroprevalence of SARS-CoV-2 in 1186 Equids Presented to a Veterinary Medical Teaching Hospital in California from 2020 to 2022"

_viruses, 2022, doi:10.3390/v14112497_

Round 1
Reviewer 1 Report
Dr. Lawton and colleagues describe a serologic study on SARS-CoV-2 in horses and associated risk factors in this manuscript.
I found this study to be very valuable because it satisfies the need for continued SARS-CoV-2 surveillance in domestic (and wild) animals for both one health and animal health objectives.
The sample size was adequate to allow for the generation of interesting hypotheses regarding the determinants connected to the seroprevalence of SARS-CoV-2.
The limitations of the study are also acknowledged correctly.
Just two minor observations for the authors:
1) I advise citing more recent in silico studies on horse ACE2 in addition to recent publications to highlight the growing body of knowledge about animal hosts.
2) I wouldn't generalize about seasons being a risk factor in such a way.
In fact, if I correctly understand the 2021 breaks, only 2020 samples were taken throughout each of the four seasons, and the sampling was completed in March 2022.
The greater number of seropositives in spring 2022, however, is still intriguing and emphasizes the significance of researching the potential expansion of the viral host range.
Thanks for this study
Best regards,
Author Response
Reviewer 1
1) I advise citing more recent in silico studies on horse ACE2 in addition to recent publications to highlight the growing body of knowledge about animal hosts.
The investigators have added recent in silico studies on horse ACE-2.
2) I wouldn't generalize about seasons being a risk factor in such a way.
In fact, if I correctly understand the 2021 breaks, only 2020 samples were taken throughout each of the four seasons, and the sampling was completed in March 2022. The greater number of seropositives in spring 2022, however, is still intriguing and emphasizes the significance of researching the potential expansion of the viral host range.
The investigators agree with reviewer that the greater number of seropositive horses detected in the spring time may relate to a possible expansion of the viral host range. A comment was added in the discussion.
Reviewer 2 Report
This is a nice, neat, well-written study about the role domestic horses play in SARS-CoV-2 circulation. I have just a few comments:
Methods: As the ELISA methods are almost identical to those described in Lawton et al. 2022 (ref. 7 in this manuscript) it would be appropriate to reduce this section considerably by just stating 'as described in Lawton et al.' and then to note any differences.
Results: It is stated that more samples were submitted in winter. Is it worth speculating why this might be?
Results: As most human infections occur in winter & spring, but statistically more samples from horses tested positive in spring, could this reflect a time-lag related to horses acquiring their infections from humans?
Discussion - line 196: change to '...more common for the elderly...'
Conclusion - line 250: add '...paralleled the seasonal peak of COVID-19 in human populations.'
Author Response
Reviewer 2
1) Methods: As the ELISA methods are almost identical to those described in Lawton et al. 2022 (ref. 7 in this manuscript) it would be appropriate to reduce this section considerably by just stating 'as described in Lawton et al.' and then to note any differences.
As suggested by the reviewer, the method has been shortened and referred to Lawton et al., 2022.
2) Results: It is stated that more samples were submitted in winter. Is it worth speculating why this might be?
Although more samples were collected/submitted during the winter months, the difference was not significant.
3) Results: As most human infections occur in winter & spring, but statistically more samples from horses tested positive in spring, could this reflect a time-lag related to horses acquiring their infections from humans?
This is an interesting observation, although we are missing the information pertaining to the COVID-19 status of owners and care takers of the horses. Further, seroconversion observed in a handful of study horses took as little as 13 days in one horse.
4) Discussion - line 196: change to '...more common for the elderly...'
The sentence has been changed.
5) Conclusion - line 250: add '...paralleled the seasonal peak of COVID-19 in human populations.'
The sentence has been changed.